# Metal Complex Formation and Anticancer Activity of Cu(I) and Cu(II) Complexes with Metformin

**DOI:** 10.3390/molecules26164730

**Published:** 2021-08-05

**Authors:** Sherin Abdelrahman, Mawadda Alghrably, Marcello Campagna, Charlotte Armgard Emma Hauser, Mariusz Jaremko, Joanna Izabela Lachowicz

**Affiliations:** 1Laboratory for Nanomedicine, Division of Biological and Environmental Science and Engineering, King Abdullah University of Science and Technology, Thuwal 23955-6900, Saudi Arabia; sherin.abdelrahman@kaust.edu.sa; 2Division of Biological and Environmental Sciences and Engineering (BESE), King Abdullah University of Science and Technology (KAUST), Thuwal 23955-6900, Saudi Arabia; mawadda.alghrably@kaust.edu.sa; 3Department of Medical Sciences and Public Health, University of Cagliari, Policlinico Universitario, 09124 Cagliari, Italy; mcampagna@unica.it

**Keywords:** metformin, copper complexes, diabetes, cancer, proliferation, protonation constants, stability constants, glutathione, NMR

## Abstract

Metformin has been used for decades in millions of type 2 diabetes mellitus patients. In this time, correlations between metformin use and the occurrence of other disorders have been noted, as well as unpredictable metformin side effects. Diabetes is a significant cancer risk factor, but unexpectedly, metformin-treated diabetic patients have lower cancer incidence. Here, we show that metformin forms stable complexes with copper (II) ions. Both copper(I)/metformin and copper(II)/metformin complexes form adducts with glutathione, the main intracellular antioxidative peptide, found at high levels in cancer cells. Metformin reduces cell number and viability in SW1222 and K562 cells, as well as in K562-200 multidrug-resistant cells. Notably, the antiproliferative effect of metformin is enhanced in the presence of copper ions.

## 1. Introduction

Metformin (*N*,*N*-dimethylbiguanide; 3-(diaminomethylidene)-1,1-dimethylguanidine) is an oral drug used daily as a first line treatment for type 2 diabetes mellitus (T2DM) by at least 120 million patients worldwide [1]. Different mechanisms of metformin action have been proposed at the cellular level. These include: the suppression of mitochondrial respiration by inhibition of complex I [2]; the regulation of AMP-activated protein kinase (AMPK) and the mechanistic target of rapamycin complex 1 (mTORC1) by multiple, mutually nonexclusive mechanisms (independent from the inhibition of ETC and cellular ATP level) [3]; and interaction with mitochondrial copper ions, which may suppress mitochondrial function [4]. Nevertheless, the exact mechanisms of metformin therapeutic action are still poorly understood.

Metformin and other biguanide derivatives were developed in the 1950s after the finding that the blood glucose-lowering ingredient in goat’s rue is guanidine. Guanidine and biguanide lead to liver damage, while metformin is a derivative with lower toxicity. In half a century of use, numerous metformin side-effects have been noted [5].

Numerous observational studies show the co-occurrence of cancer and diabetes [6]. It is widely accepted that diabetes is a significant risk factor for different cancer types, regardless of its connection to obesity [7,8,9]. The strongest correlation between diabetes and cancer correlation has been observed in liver, breast, pancreas, colorectum, and endometrial tumors. Metformin-treated diabetic patients seem to have a lower risk of cancer than those who use other therapeutics [10,11]. Since diabetes is an independent cancer risk factor, diabetes treatment might reduce cancer risk [12]. However, it is still unclear if the cancer-suppressing role of metformin is related to a direct preventive effect of the therapy or the cancer–diabetes connection [12]. 

The association between metformin use and reduced cancer risk in patients with diabetes was suggested in a pioneering observational study published in 2005, which reported a 23% decrease in cancer risk with metformin use [13]. Since then, several epidemiologic studies have provided additional evidence of lower cancer risk in diabetic patients treated with metformin than in non-metformin subjects. Along with epidemiologic studies, several meta-analyses have supported metformin use and reduced cancer risk in diabetic patients with cancer [14,15]. In particular, reduced incidence of liver [16], pancreatic [17], colorectal [18], breast [19], kidney [20], and lung cancer [21] have been associated with metformin use. In addition, metformin was shown to be an effective adjuvant drug for cancer patients, and could help patients with colorectal and prostate tumors who are undergoing radical radiation therapy [22]. Several studies have shown recently that metformin inhibits cancer invasion and migration, which could account for the improvement of prognosis in cancer patients [23,24]. 

In recent years, interactions between either metformin or metformin complexed with copper (copper/metformin) and nucleic acids have been investigated. Shahabadi et al. [25] have shown that free metformin forms stable adducts with calf thymus DNA (CT-DNA; K_b_ = 8.3 × 10^4^ M^−1^) [25]. Shoair et al. [26] additionally proved that copper(II)/metformin complexes bind to DNA through an intercalative mechanism. Moreover, stable Cu(II)/metformin/DNA adducts had antioxidant and biomimetic catalytic activities in in vitro tests. 

Copper is a redox active metal and its chelates can be used in therapy as catalytic metallodrugs. Possible catalytic reactions are: transfer hydrogenation, C-C bond cross-coupling, and bond cleavage by hydrolysis or oxidation, among others [27]. For instance, copper complexes with thiosemicarbazone derivatives leads to an oxidative cell damage in human melanoma cells and oxidative damages of DNA, which arrest G2/M cell cycle [28]. Copper(II) complexes with 4-Chloro-3-Nitrobenzoic acid ligand intercalatively bound DNA, and have and antiproliferative effect in human cancer cell lines, by inhibiting G0/G1-phase cells and leading to apoptosis [29].

In order to understand the activity of metformin and its copper complexes at the cellular level, we studied Cu(II) complex formation with metformin. We present coordination studies of copper (I)/metformin and copper (II)/metformin complexes in relation to glutathione (GSH), which is a representative copper-binding peptide that exists at high levels in cancer cells. Moreover, we show the effect of metformin and its copper complexes on the proliferation, morphology, and viability of human colorectal adenocarcinoma, chronic myelogenous leukemia, and adriamycin-resistant chronic myelogenous leukemia cells. 

## 2. Materials and Methods

### 2.1. Reagents

Metformin, glutathione, D_2_O, NaOD, DCl, CuCl, CuCl_2_, CuSO_4_, NaOH, and HCl were purchased from Sigma-Aldrich (Milano, Italy) and used without further purification.

A previously described method was used in the preparation of 0.1 M carbonate-free KOH solution [11]. Metformin solutions were acidified with stoichiometric equivalents of HCl. Metal solutions were prepared by dissolving the required amount of metal salt in pure double distilled water containing a stoichiometric amount of HCl to prevent hydrolysis. These solutions were standardized by the complexometric method with EDTA and proper indicators.

### 2.2. Potentiometry-UV-Vis

Protonation and complex-formation equilibrium studies were carried out under the same conditions described in previous publications [30,31]. The operating ligand concentrations ranged from 5 × 10^−4^ to 4 × 10^−3^ M. Complex formation studies were carried out using constant ligand concentration, and 1:1, 1:2, and 1:3 metal/ligand molar ratios. Combined potentiometric–spectrophotometric measurements were performed for protonation equilibria in the 200–400 nm spectral range and in the 400–800 nm spectral range for copper (II) complexes, using 0.2 and 1 cm path lengths, respectively. Protonation and complex formation data were analyzed using the Hyperquad and HypSpec2014 program [32].

### 2.3. Nuclear Magnetic Resonance (NMR) Spectroscopy

^1^H-NMR spectra were recorded for metformin and glutathione in combination with CuCl and CuCl_2_. Metformin was dissolved in 800 μL at a 1:9 ratio of D_2_O:H_2_O, to make a 5 mM metformin solution. The pH of the solution was manually adjusted to the desired pH (ranging from 2 to 12) with a 0.1 M NaOD and 0.1 DCl stock solution in 100% D_2_O. All NMR experiments were performed on the 600 MHz Bruker spectrometer at 25 °C (Karlsruhe, Germany). The NMR data were processed and analyzed by TopSpin software (4.1.3, Bruker, Karlsruhe, Germany).

The Cu(II) and Cu(I) measurements were prepared through the stepwise addition of a proper copper ion stock solution (0.01 M in 100% H_2_O for the samples in a 90:10 (*v*/*v*) H_2_O/D_2_O mixture) to the solution of metformin, giving metformin:Cu(II) and metformin:Cu(I) molar ratios of 1:0, 1:0.02, and 1:0.05 (for Cu(II) ions) and 1:0.5 (for Cu(I) ions). For each metformin:Cu(II) and metformin:Cu(I) ratio, the pH was controlled and adjusted to the range from 4 to 9 with 0.1 M NaOD and 0.1 M HCl stock solutions. The same procedure was applied to glutathione, copper ion, and metformin experiments.

### 2.4. Cellular Studies

#### 2.4.1. Cell Culture of SW1222, K562, and K562-200

Three cancer cell lines were used: SW1222, human colorectal adenocarcinoma (Merck, Darmstadt, Germany); K562, chronic myelogenous leukemia; and an adriamycin-resistant CML cell (K562/Adr200). K562/Adr200 is a recently developed cell line from the center of excellence in genomic medicine research, KAU, which exhibits multidrug resistance [33].

SW1222 were cultured in DMEM (Gibco, New York, NY, USA) media supplemented with 10% fetal bovine serum (FBS) (Gibco, New York, NY, USA) and 1% penicillin/streptomycin (Gibco, USA). Both K562 and K562/Adr200 were cultured in RPMI media (Gibco, New York, NY, USA) supplemented with 10% FBS and 1% penicillin/streptomycin. The seeding density for all the three cell lines was 1000 cells per well in 96 wells plates. 

The following concentrations of metformin were used in media at DIV0: 20 μM, 1 mM, 5 mM, 10 mM, 15 mM, and 50 mM. From this experiment, the effective concentration that inhibited the growth of ≥50% of the cells was determined. In the subsequent experiments, copper sulfate (CuSO_4_) was added to cells with and without metformin at 5 μM, 100 μM, 400 μM, and 5 mM concentrations, to assess its effect on cells treated with or without metformin. Both metformin and copper sulfate were prepared in sterile nuclease-free water, then added to the media. The cells were maintained in culture for 72 h before any further assessments were done.

#### 2.4.2. Assessment of the Effect of Metformin on the Proliferation of Cancer Cell Lines in the Presence and Absence of Copper Sulfate

To determine the effective concentration of metformin that inhibits the growth of more than 50% of each of the three cancer cell lines, ATP production was assessed 72 h after drug addition. A control sample of untreated cells was included in all experiments. The half-maximal inhibitory concentration50 (IC_50_) of metformin was then calculated using Prism 9 software (8.0.0, GraphPad Prism, San Diego, CA, USA). Selected concentrations of the drug that inhibited 50% or more of the cells were then used in the subsequent experiments (i.e., 10, 15, and 50 mM). In the following experiments, copper sulfate (CuSO_4_) was added in varying concentrations (5, 100, 400 µM, and 5 mM) to non-treated and treated cells with metformin. The CellTiter-Glo^®^ luminescent cell viability assay was used to determine levels of ATP release from each cell line after 72 h, according to manufacturer’s instructions. ATP release was compared between cells treated with metformin and non-treated cells. In subsequent experiments, the ATP release was compared between metformin-treated cells in the absence or presence of CuSO_4_. The ATP release of cells treated with CuSO_4_ in the absence of metformin was also assessed, as a control. 

Cell viability was also assessed by using the live/dead viability/cytotoxicity kit for mammalian cells, commercially available from Thermo Fisher Scientific (Waltham, MA, USA). This assay relies on intracellular esterase activity in live cells that changes Calcein into Calcein-AM, which then emits green fluorescence. Additionally, the kit assesses the integrity of the plasma membrane. Ethidium homodimer-1 is internalized through damaged membranes, leading to red fluorescence in dead cells that are detected by fluorescence imaging.

#### 2.4.3. Immunostaining of Actin Stress Fibers

To assess the effects on actin stress fiber organization and cell morphology, rhodamine phalloidin was used. Cells were fixed in 4% paraformaldehyde 72 h post-treatment. Cells were permeabilized for 5 min in prechilled 0.5% Triton × 100, 3 mM MgCl_2_, and 300 mM sucrose, followed by washing with 1× DPBS. A blocking buffer composed of 5% fetal bovine serum (FBS), 0.1% Tween 20, and 0.02% sodium azide was then added to the cells for 30 min. Rhodamine phalloidin was incubated for one hour after removing and discarding the blocking buffer. Cells were subsequently washed with 1× PBS, and DAPI was added for nucleic acid staining prior to fluorescence imaging.

## 3. Results and Discussion

### 3.1. Metformin Protonation Constants

The electronic structure of drugs determines their physicochemical properties, including solubility and charge. This in turn determines their mechanism of action and, thus, biological activity. Understanding drug structure helps to predict and understand bioavailability, membrane interactions, and mechanisms of action in biological systems.

Metformin, hereafter referred also as Ligand (L), is a highly water-soluble molecule (1.06 × 10^6^ mg/L (miscible) at 25 °C [34]), but its physicochemical properties in water are poorly understood. The pK_a_ value and hydrophilicity of a molecule plays an important role in its cellular activity. Protonation constants of metformin in water have been studied by different techniques—such as potentiometry and NMR—giving distinct results (some literature examples are shown in Table 1). Structural dynamics of protonated metformin forms were recently also investigated by Raman scattering and theoretical calculations [35].

Here, we use ^1^H-NMR titration (Figure 1) and combined potentiometric-UV titration (Figure 3) in changing pH conditions. Protonation constants were determined on the basis of potentiometric-UV titration (Table 1), while ^1^H-NMR data revealed structural changes in metformin upon deprotonation of nitrogen atoms. At pH 2, metformin is fully protonated and the overall charge is equal to 3 (Figure 2). First dissociation occurs at acidic pH and leads to a shift of N_B_, N_D_, and N_E_ nitrogen (Figure 1A), as well as methyl protons (Figure 1B). Of note, nitrogen protons can be observed in the condition of a 1:9 ratio of D_2_O:H_2_O, and the intensity of the N_E_ proton signal increases above pH 3.11, while the N_B_ and N_D_ proton decreases. Moreover, the first dissociation leads to significant UV spectra changes (Figure 3).

**Figure 1 molecules-26-04730-f001:**
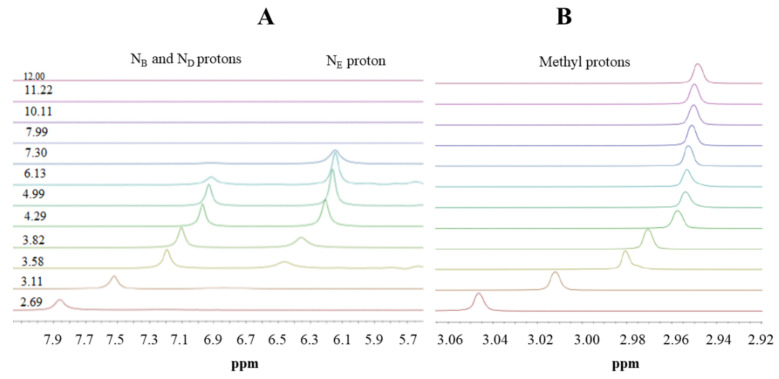
^1^H-NMR spectra of metformin in different pD conditions. (**A**) 6.7–8.2 ppm range; (**B**) 2.9–3.0 ppm range. Proton chemical shifts estimation is presented in Appendix A.

The second deprotonation occurs at slightly basic pH (Figure 2, Table 1) and leads to the disappearance of nitrogen protons in the ^1^H-NMR spectrum (Figure 1A), while there are no significant changes in the UV spectrum (Figure 3). Conversely, the last deprotonation occurs at highly basic pH (Figure 2, Table 1) and lead to slight changes in the UV spectra ~200 nm (Figure 3), while no changes can be observed in the ^1^H-NMR spectrum (Figure 1).

**Figure 2 molecules-26-04730-f002:**
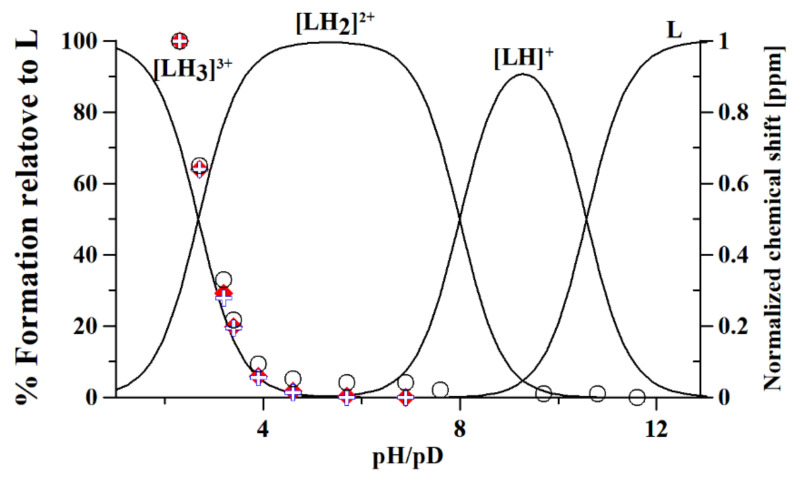
Speciation plot of metformin (L), calculated on the basis of protonation constants reported in Table 1. The normalized chemical shifts of each proton overlap the speciation plots: (-NH) + and ♦; (-CH_3_) ○; protons (Scheme 1).

**Scheme 1 molecules-26-04730-sch001:**
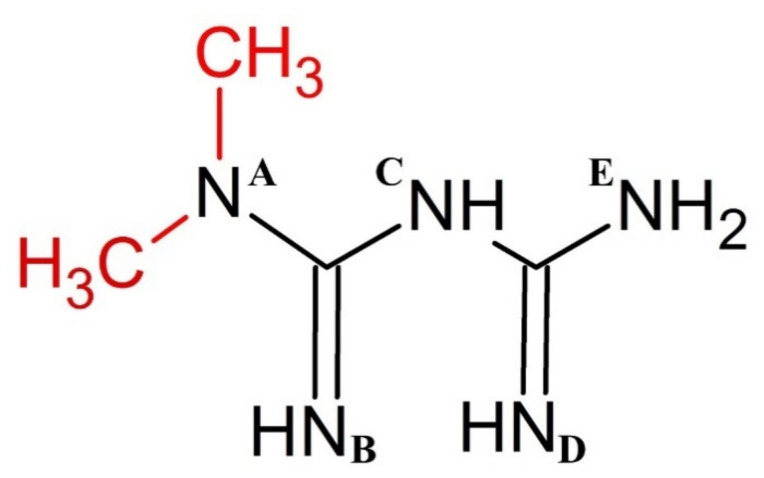
Chemical structure of metformin (biguanide backbone shown in black). Letters A–E denote nitrogen atoms that are referred to in the discussion section.

**Figure 3 molecules-26-04730-f003:**
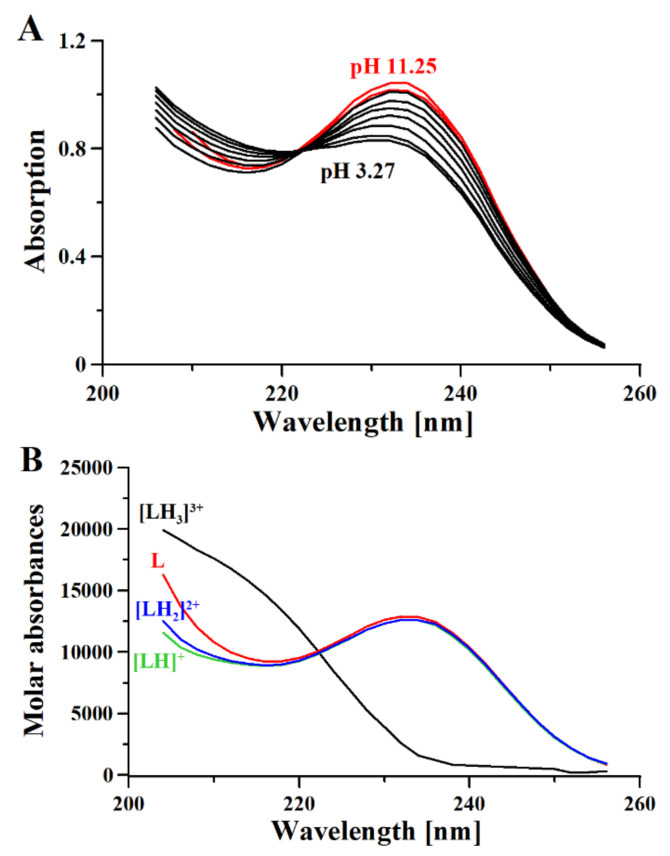
UV spectra (**A**) and Molar absorptivity spectra (**B**) of metformin (L) calculated with HysSpec2014 [32]. [L] 0.4 mM; path length 0.2 cm.

**Table 1 molecules-26-04730-t001:** Protonation constants of metformin at 25 °C, 0.1 M NaCl ionic strength, obtained using the HypSpec2014 [32] program for combined potentiometric and spectrophotometric measurements and HypNMR [36] for ^1^H-NMR measurements. NA = not applicable due to insufficient data.

Technique/Species	Experimental Data			Literature Data		
	LH_3_	LH_2_	LH	LH_3_	LH_2_	LH
Potentiometry	2.67(2)	8.00(1)	10.57(1)	-	2.79 [37]	11.02 [37]
				-	2.90 [38]	11.10 [38]
UV				-	-	-
NMR	2.8(1)	NA	NA	-	3.1 [39]	13.8 [39]

Based on the results presented here, a new metformin dissociation model can be proposed (Scheme 2). The lowest pK_a_ (2.67) can be attributed to the N_A_ proton, which leads to a shift of adjacent methyl protons and N_B_ and N_D_ protons (Figure 1). A further increase in pH leads to the second dissociation with a pK_a_ value of 8.00, which could be assigned to the proton of N_B_, N_C_, or N_D_. The exact proton position is hard to ascertain, and the interconversion of different biprotonated forms may occur, and/or the proton is shared between N_B_ and N_D_ atoms in the pH range of 3.6–7.3 (Figure 1). The last dissociation, with a pK_a_ value of 10.57, can be attributed to the terminal amine group (N_E_) and leads to the loss of positive charge. The change of charge from +1 to 0 leads to minor changes in the UV spectra (Figure 3), and no proton chemical shift can be observed due to absence of adjacent protons or hydrogen bonds stabilizing proton positions in our ^1^H-NMR experimental conditions. 

Previous literature reports only two protonation constants for metformin, although Raman spectra recorded at pH 7 (and pD 7.4 in heavy water) are distinct from those recorded at very acidic and basic pH [35]. Importantly, Raman scattering with theoretical calculations by Hernandez et al. [35] supports the hypothesis of N_B_ and N_D_ protonation with proton sharing through intramolecular N-H^…^N hydrogen bonds (Scheme 2). Such intramolecular hydrogen bonds could also explain the unusual presence of a nitrogen proton signal in ^1^H-NMR in the condition of a 1:9 ratio of D_2_O:H_2_O. As shown by Bharatam et al. [40], due to the presence of protic hydrogen, the most stable neutral form of metformin in water is a tautomer in which protons of the N_C_ group are formally transformed to N_B_, N_D_, or N_E._ In addition, the presence of a delocalized π-electron system leads to a quasiplanar structure of protonated forms, which could be further stabilized in biological systems by interactions with aromatic residues of proteins/peptides and nucleic acids [35].

The determination of third protonation constant significantly influences the deprotonation scheme of Metf in changing pH conditions, and further influences the metal complex stability. Moreover, the presented here deprotonation sequence supports the previous findings of Metf planar structure at physiological pH, which facilitates the Metf interaction with biologically important molecules. 

### 3.2. Copper Complexes

The biochemical and therapeutic activity of metformin is related to its electron distribution. For instance, the highly basic nature of metformin is due to electron delocalization [40]. Metformin, as with other biguanide derivatives, forms complexes with transition metal ions (including copper (II) ions) [41], due to the localization of charge density on the terminal nitrogens, which enhance the stability of the formed chelates [40].

Combined potentiometric-Vis titration in changing pH conditions (Figure 4) was used to determine Cu(II)/metformin complex stoichiometry and cumulative stability constants (Table 2), while ESI-MS measurements (Appendix A) were used to establish the stoichiometry of Cu(II)/metformin complexes. Moreover, Cu(II)/metformin ^1^H-NMR titration with growing metal-to-ligand molar ratio (Figure 5) and in changing pH conditions were used to establish metal-binding sites in the formed chelate. 

The potentiometric-Vis data show that [CuLH]^3+^ starts to form above pH 2 (Figure 4A), while the formation of [CuL_2_H]^3+^ complexes above pH 4 leads to a hypsochromic shift in the Vis spectrum (Figure 4B,C) due to the N(amine) → Cu(II) transition. [CuL_2_]^2+^ is formed above pH 5, after the dissociation of protons, and leads to hypsochromic and hyperchromic shifts (Figure 4B,C) due to additional nitrogen participation in the metal coordination core (the N(amine) → Cu(II) transition). The ESI-MS spectrum (Appendix A) showed the formation of [CuLH_−1_]^+^ (C_4_H_10_N_5_Cu; 191.019 *m*/*z*), [CuL]^+^ (C_4_H_11_N_5_Cu; 192.026 *m*/*z*), and [CuLCl]^+^ (C_4_H_11_N_5_CuCl; 192.026 *m*/*z*) complexes in the mass spectrometry conditions (pH 4; 1:1 H_2_O:MetOH).

**Table 2 molecules-26-04730-t002:** Experimental metformin/Cu^2+^ complex cumulative formation constants at 0.1 M KCl ionic strength and 25.0 °C calculated with HypSpec2014 [32] program for combined potentiometric-spectrophotometric measurements. [L] = 4 mM; [Cu^2+^] = 2 mM. The experimental data are compared with the literature data.

Technique/Species	Logβ (Experimental Data)			LogK (Literature Data)	
	CuLH	CuL_2_H	CuL_2_	CuL	CuL_2_
Potentiometry	17.87(5)	28.3(1)	22.20(7)	7.17 [42]	5.13 [42]
				8.59 [38]	6.35 [38]
Vis				-	-

**Figure 4 molecules-26-04730-f004:**
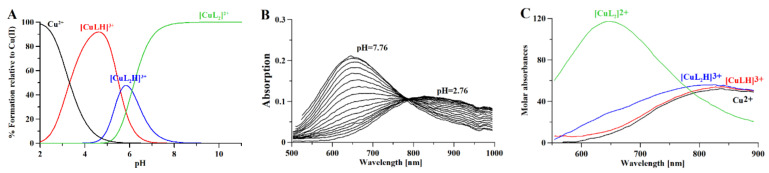
(**A**) Distribution curves for Cooper (II) complexes with metformin calculated on the basis of stability constants (Table 2) with the Hyss program [43]; (**B**) Vis spectra of Cooper (II) complexes with metformin and (**C**) molar absorptivity spectra calculated with HypSpec2014 [32]. [L] = 4 mM; [Cu^2+^] = 2 mM; path length 1 cm.

**Figure 5 molecules-26-04730-f005:**
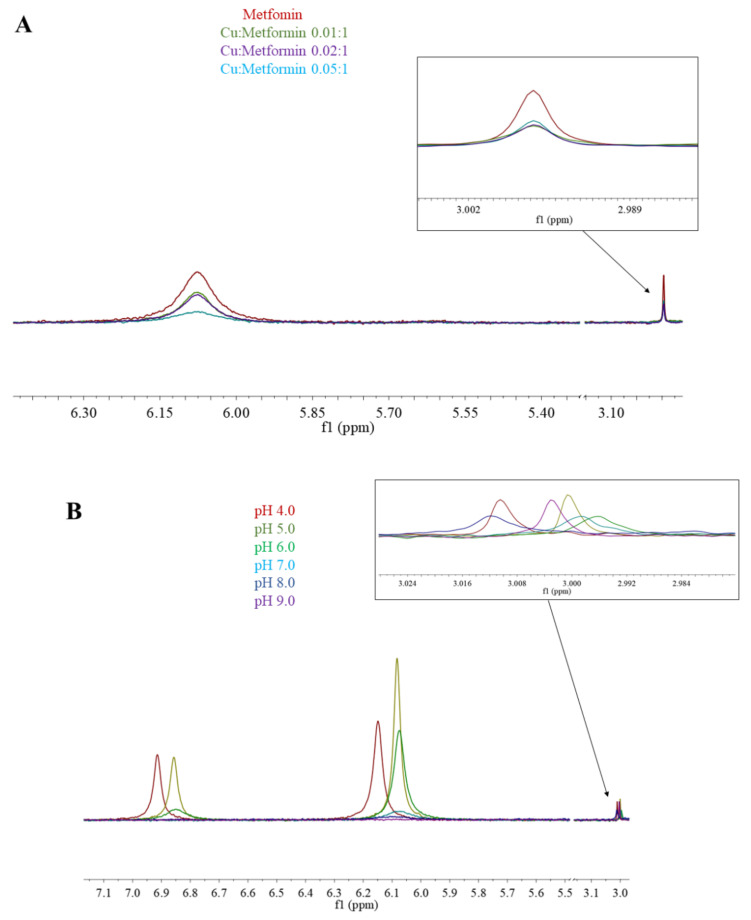
^1^H-NMR spectra of (**A**) Cu(II)/metformin titration; pH 7.4; (**B**) Cu(II)/metformin (0.05:1) pH titration.

In the ^1^H-NMR experiments, the addition of paramagnetic Cu(II) to the solution caused signal broadening due to the paramagnetic contribution of Cu(II) to the spin-lattice relaxation (R1) rates of H protons. Figure 5A shows the results of metformin titration with Cu(II) ions at fixed pH (7.4), while Figure 5B presents the chemical shifts of Cu(II)/metformin complex protons in the pH range 4–9. The ^1^H-NMR spectrum of metformin at pH 7.4 (Figure 1 and Figure 5A) shows two signals, which can be attributed to N_E_ proton and methyl group protons. The addition of copper (II) ions to the metformin solution leads to a broadening of the signal, suggesting complex formation in the vicinity of protons. 

At pH 4.0, nearly 100% of [CuLH]^3+^ complex is formed (Figure 4A) and in the ^1^H-NMR spectrum of Cu(II)/metformin solution (Figure 5B), two signals attributed to N_B_, N_D_ and N_E_ protons and one signal from methyl groups can be observed. At pH 5.0, all tree signals are shifted upfield, while a further pH increase leads to drastic lowering of amine proton signals and upfield shift of methyl protons. At pH 9.0 only the methyl protons can be observed. Of note, the potentiometric-Vis titration does not show changes in the complex stoichiometry in the pH range 7–9 (Figure 4), while ^1^H-NMR spectra present proton translocation (Figure 5). Such phenomenon was previously studied by X-ray crystalography of Cu(II)/metformin complexes in different pH conditions [44,45].

Among five known X-ray structures of Cu(II)/metformin, three include neutral metformin (CSD code: AJUHUJ, HIBPOX, and HIHDUX; Appendix A) and the other two structures were made under basic conditions (CSD code: EFIXUM and ETOFOI), where N_C_ has no proton (Appendix A). Of note, structures AJUHUJ, HIBPOX, and HIHDUX are perfectly planar, while in EFIXUM and ETOFOI structures, the planarity is perturbated.

It remains uncertain if metformin can bind Cu(I) and form adducts with biologically important molecules such as peptides. In water, Cu(II) is more stable than Cu(I), while in living organisms, Cu(I) is tightly bound by specific proteins to avoid potentially damaging redox reactions. It is likely that metformin will be forced to interact with protein/amino acid-bound copper ions, in particular with copper (I), and compromise the vast majority of intracellular copper ions [2]. As shown by Repiscak et al. [2], N_B_ and N_D_ atoms have sp^2^-hybridization, similar to that of imidazole in histidine, which means that π-backbonding can occur and stabilize the oxidation state of Cu(I) by transferring electron density into the π-orbitals of the ligand.

In a recent study of the antimicrobial activity of Cu(II)/metformin complexes, Olar et al. [46,47] showed with cyclic voltammogram a [Cu^II^(metformin)_2_]^2+^ reduction peak (against Ag/AgCl electrode) at 320 mV and an oxidation peak at 490 mV, giving an E_1/2_ of 405 mV, which lies in the range of the reduction potentials of copper-containing enzymes [48].

Previous theoretical calculation studies by Repiscak et al. [2] showed that metformin can form stable complexes both with Cu(II) and Cu(I). Importantly, the Cu(II) complexes are more than 200 kcal/mol more stable than Cu(I) complexes due to stronger Coulomb and orbital interactions.

GSH (Scheme 3) is a tripeptide composed of γ-glutamic acid, cysteine, and glycine, with an intracellular concentration of 1–10 mM in mammalian cells [49]. Under normal conditions, the majority of cellular GSH is present in a reduced form. Oxidation into GSSG occurs under direct interaction with free radicals. GSH metabolism has a complicated role in both cancer and antineoplastic therapy. While GSH is important in the detoxification of carcinogens, its elevated state in many types of tumor may also increase resistance to chemo- and radiotherapy [50,51,52,53].

Although metformin-metal complexes have been studied with numerous analytical techniques, including NMR, EPR, and IR spectrometry and voltammetry, this work used free copper ions, while in biological systems, copper ions are bound to proteins or peptides. In our studies, we investigated complex formation of GSH both with Cu(I) and Cu(II), and studied the interaction of the chelate with metformin (Figure 6). The characteristic feature in the ^1^H-NMR spectrum is the broadness of each signal upon addition of copper ions. Cu(I) is a diamagnetic ion, and signal broadening is indicative of a slowing down of the motion of the complex solution, probably due to the formation of polymeric species (GS^−^-Cu(I)) and/or the exchange between complexes of differing stoichiometries [54].

In solutions containing GSH and Cu(I) ions (Figure 5A), all signals of GSH are lowered and additional signals (*) at ~3.5 and ~1.0 ppm appear, which show Cu(I)/GSH complex formation. The addition of metformin to the complex solution leads to the lowering of all signals and appearance of signal (♦) at 2.88, which can be associated to free metformin.

The addition of Cu(II) ions into the solution of GSH leads to the broadening of all signals and the appearance of new signals (*) at ~7.98 and ~2.98 ppm, suggesting metal complex formation. The addition of metformin to the complex solution leads to the lowering of all signals and appearance of signals (♦) at 2.99 and 2.88 ppm, which can be attributed to metformin complexed with copper and free metformin, respectively.

There is only one crystal structure of oxidized GSSG with copper (II) ions [55], where copper (II) ions are coordinated by two deprotonated peptide nitrogens, the glutamic amine nitrogen, and the glycinyl terminal carboxylate oxygen in an approximate planar coordination, while the cysteinyl sulfur is bonded apically to form a square pyramid. In our experimental conditions, glutathione is present in the reduced form (GSH) and the ^1^H-NMR chemical shift pattern for Cu(II) complex (Figure 6B) supports the coordination scheme presented by Miyoshi et al. in the solid state [55]. The chemical shift pattern for Cu(I) complex (Figure 6A) is slightly different and suggests greater involvement of the glutamic amine nitrogen and deprotonated glycine peptide nitrogen. The signal lowering of the copper/GSH complexes upon addition of Metf suggest the formation of ternary Metf/copper/GSH complexes, but the exact structure of such adducts needs to be further elucidated with additional studies.

The presented results show the formation of stable Cu(II)/metformin complexes, which can form adducts with GSH, one of the main peptides involved in metal intoxication and antioxidative molecular mechanism. The metal complex formation of Metf and interaction of copper/Metf complex with biologically active molecules could interfere with regular molecular process in the cell, and lead to cell death.

### 3.3. Metformin and Copper Anticancer Activity

Metformin has been found to exhibit anticancer activity against various tumor types. Its activity may relate to either targeting of the adenosine monophosphate-activated protein kinase (AMPK) pathway with subsequent inhibition of mTOR signaling, or through reducing insulinemia and glycemia. Copper is usually found in very high concentrations in cancer tissues, since it promotes angiogenesis and many other important biological processes.

In recent studies, Lu et al. [56] showed that metformin is cytotoxic to human gastric cancer AGS cells. Their results demonstrated that metformin significantly reduced cell viability at 20 mM for 12 h, in a concentration- and time-dependent manner. Metformin-treated AGS cells also had morphological alterations, such as cell shrinkage, nuclear condensation, membrane blebbing, and rounding. Thus, metformin suppressed AGS cell growth via induction of apoptotic death. Additionally, Lu et al. [56] demonstrated that metformin has no significant effect on the viability of normal colon CCD 841 CoN cells and embryonic lung HEL 299 cells and 293 cells when used in a concentration range of 10–50 mM for 48 h. Contrasting data are also reported in literature, however [5].

In our study, we assessed the anticancer properties of metformin in vitro against colorectal cancer (SW1222) and leukemia (K562) cell lines, and studied the effect of copper (II) ions on the anticancer activity. Furthermore, we aimed to explore the potential of metformin and its copper complexes as an anticancer agent, using the K562/Adr200 leukemic cell line, which exhibits multidrug resistance (MDR)—a major challenge in chemotherapy.

Human colorectal adenocarcinoma (SW1222) (Merck, Germany), chronic myelogenous leukemia (K562), and the adriamycin-resistant K562/Adr200 SW1222 cell line were treated with 20 µM, 1 mM, 5 mM, 10 mM, 15 mM, and 50 mM of metformin [56]. Effects were assessed after 72 h of culture. Immunostaining of SW1222 actin fibers and nuclei showed shrinkage and rounding of cells as the concentration of metformin was increased (Figure 7A,C). At 15 mM and 50 mM metformin, cell shrinkage and rounding was dramatically increased (Figure 7B,C). Moreover, metformin induced the loss of stress fibers in a large number of cells at the same concentrations (Figure 8C). Interestingly, both K562 and K562-200 were more sensitive to the effect of metformin, showing a loss of actin stress fibers at concentrations as low as 1 mM and 20 µM sequentially (Figure 7A, Appendix A). Notably, 20 µM is close to the plasma concentration of metformin in T2DM patients [57]. Cell viability was measured by live/dead assay, measuring the luminescent signal, which is proportional to ATP produced in viable cells. ATP from untreated SW1222 cells was compared to cells treated with various doses of metformin. A significant difference was found at 5, 10, 15 mM, and 50 mM (*p* values of *p* = 0.04, *p* = 0.05, *p* = 0.02 and, *p* = 0.006, respectively) (Figure 7C). Similarly, K562 cells were also sensitive to metformin, with significant differences in the amount of ATP produced when comparing treated and non-treated populations (Figure 7C). Interestingly, the effect of metformin on the MDR leukemic cell line K562-200 was the strongest, as shown by the drastic inhibition of the ATP release as metformin concentration increases. The *P* values were found to be *p* = 0.006 at 20 µM and *p* < 0.001 at all the remaining metformin concentrations (Figure 7C). The IC_50_ for metformin was accordingly calculated and found to be 15.6 mM for SW1222, 851.2 µM for K562, and 63.1 µM for K562-200 (Figure 7B). Live/Dead staining of cells in the three cell lines showed a concentration dependent decrease in the proportion of live cells, which was more evident in the MDR leukemic cell line K562-200 (Figure 8A,B, Appendix A).

Three concentrations of metformin were selected for subsequent experiments based on the observed effects on cell viability and proliferation: 10 mM, 15 mM, and 50 mM. To test the effect of Cu(II) on the properties of metformin (Figure 9), copper (II) sulfate was used in the following concentrations: 5 µM, 100 µM, 400 µM, and 5 mM [58]. To identify the causative factor for the observed effect on the cells at every concentration, each experiment included an untreated controls for both metformin and Cu(II).

In 5 mM Cu(II), almost all cells died, regardless of metformin concentration. In the absence of metformin, Cu(II) had only a slight negative impact on cell proliferation at 5 µM and 100 µM. However, at 400 µM, Cu(II) led to the inhibition of cell proliferation in ≥50% of the cells in the three cell lines. Addition of 5 µM and 100 µM Cu(II) combined did not have any noticeable effect on the properties of metformin, regardless of the concentration. However, when 400 µM of Cu(II) was added with 10 mM metformin, the resulting inhibition of cell proliferation was double that caused by metformin alone. This was similar to the inhibition caused by 400 µM Cu(II) in the absence of metformin. An exception was K562, where the normalized ATP percentages were 35.2% at 10 mM metformin, 47.2% at 400 µM Cu(II), and 22.5% in metformin and Cu(II) combined at the same concentrations. Furthermore, at 400 µM of Cu(II), inhibition of cell proliferation caused by 15 mM metformin was increased by >40% in the three cell lines in comparison to metformin alone. In SW1222, inhibition from metformin and Cu(II) combined was similar to that of the Cu(II) alone. Interestingly, both of the leukemic cell lines did not grow as well in metformin and Cu(II) combined than either alone. At 50 mM of metformin, the addition of 400 µM copper (II) sulfate led to a considerable increase in the inhibition rate in SW1222, K562, and K562-200 cell lines by 80%, 66%, and 39%, respectively, when compared to 50 mM metformin in the absence of copper and 79%, 85.6%, and 77.6%, respectively, when compared to copper in the absence of metformin. These results were confirmed by observations from the fluorescent immunostaining.

Our results suggest that metformin has anticancer potential against human colorectal adenocarcinoma (SW1222) and chronic myelogenous leukemia (K562) cell lines. Moreover, metformin also suppresses growth of the K562/Adr200 cell line, at concentrations that match the plasma concentration of the drug in diabetic patients (20 μM) [57]. These results confirm several previous studies showing the activity of metformin against different types of cancer [59,60,61]). The anticancer effect of metformin was found here in cell viability assays and in morphological changes observed in the cell microscopically. Moreover, we found that the sensitivity of cancer cells to metformin varies greatly depending on the cancer type [62]. The difference in the antiproliferative effect might be attributed to differences in the actual mechanism of action of metformin that leads to this outcome, as described in several studies [63,64,65].

We also assessed the effect of Cu(II) on the anticancer property of metformin. Among metals, copper is known for its role as a cofactor in many biological pathways. More importantly, copper is also found to accumulate in tumors, and due to the selective permeability of cancer cell membranes, this makes it an interesting metal in which to study the effects of its complex formation with metformin [66,67,68].

We found that Cu(II) at 400 µM led to enhanced antiproliferative effects of metformin on three cancer cell lines. This effect is similar to a previous report of copper complexes working as topoisomerase inhibitors and, thus, influencing DNA topology [69].

## 4. Conclusions

Metformin has been an economical first-line treatment in T2DM for decades. Life-long metformin treatment in millions of diabetics has highlighted the numerous side effects and correlations in the co-occurrence of other diseases, including cancer. There is growing data suggesting the anticancer effects of metformin. Our work shows that metformin has three protonation sites and can form stable complexes with copper (II) ions in water. Moreover, Cu(II)/metformin and Cu(I)/metformin form adducts with GSH, an intracellular anti-oxidative peptide, which accumulates at high levels in cancer cells.

In vitro cell cultures of the cancer cell lines SW1222, K562, and K562-200 (a multi-drug resistant form of K562) confirmed the anticancer activity of metformin on three cell lines representing two of the most common types of cancers worldwide (colorectal cancer and chronic myeloid leukemia). Additionally, we showed antiproliferative effects of metformin against K562/Adr200, an MDR leukemic cell line. Moreover, our data show that Cu(II) might enhance the antiproliferative effect of metformin on cancer cells, dependent on concentration. These findings may lead to new potential options in cancer treatment. Future studies on the effects of combining Cu(II)^−^ metformin with known chemotherapeutic agents might lead to important findings related to cancer treatment.

## Data Availability

All supplementary data are reported in electronic Appendix A.

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
