# Peer review of "Metal Complex Formation and Anticancer Activity of Cu(I) and Cu(II) Complexes with Metformin"

_molecules, 2021, doi:10.3390/molecules26164730_

Round 1
Reviewer 1 Report
The manuscript molecules-1313474 "Metal complex formation and antitumoral activity of Cu(I) and Cu(II) complexes with metformin" by Abdelrahman et.al. describes the formation of metformin based Cu(I) and Cu(II) complexes and the study of their anticancer activity on human colorectal adenocarcinoma (SW1222), chronic myelogenous leukemia (K562) and adriamycin‑resistant (K562/Adr200) cells. It was also shown that these complexes formed adducts with glutathione.
This is a very interesting and well written work. The reliability of the obtained results is beyond doubt. However, the reviewer has a number of comments / suggestions:
1) I recommend that the authors strengthen the Introduction part on the design of copper complexes. New articles on the design of copper complexes, as well as their applications, should be added. For example, Int. J. Mol. Sci. 2021, 22(6), 3104; Molecules 2021, 26(8), 2334; Molecules 2021, 26(13), 4028.
2) I suggest to add an image of the Cu-metformin and Cu-metformin-glutathione complexes and coordination of copper (II) cations.
3) In Scheme 2, under the arrows, add "– H+".
4) There was no supporting information in the attached files, so I didn't find Figures S1 and S2.
5) Line 278-279. "Such phenomenon was previously studied by X-ray crystalography of Cu(II)/metformin complexes in different pH conditions. " There is no reference. Are these the results of the authors?
6) I suggest to add a small sub-conclusion to each sub-chapter summarizing the obtained results for better understanding by the readers.
7) The manuscript lacks information on the IC50.
8) What mechanism of influence of Cu (II) on the anticancer property of metformin do the authors suggest? How can it be confirmed?
9) I suggest to change "antitumoral" to "anticancer" or "antitumor". In the text of the article, the authors use only 1 term "anticancer".
10) The manuscript should be rechecked for typos and errors. For example:
Line 127. "copper sulfate (CuCo4)" should be corrected.
Figure 6B - "GSH:Cu(I):metformin" should be corrected
Author Response
Dear Reviewer,
Together with all authors I would like to thank you for your valuable revision and suggestions regarding our manuscript, to which we all agree to. Please find in the following brief descriptions of the changes (in revised manuscript signed in red) we made in order to improve our paper, in particular with regard to data quality and presentation.
- I recommend that the authors strengthen the Introduction part on the design of copper complexes.
Reply: The manuscript is proposed for the special issue “Old drugs for new metal related diseases” and the Introduction put emphasis on the leading topis. We agree with the Referee that introduction to the Cu-complexes in the therapy of cancer is needed, and we added a new paragraph (signed in red) in the Introduction section with the reference to the latest advances (Int. J. Mol. Sci. 2021, 22(6), 3104; Molecules 2021, 26(8), 2334).
- I suggest to add an image of the Cu-metformin and Cu-metformin-glutathione complexes and coordination of copper (II) cations.
The Figure S2 represents known crystal structures of Cu(II)/Metformin, while Figure S12 (added to the revised version) shows the crystal structures of Cu(II)/GSSG. At this moment, we are not able to propose the exact structure of mixed Metf/Cu(II)/GSH adduct, while additional studies are need (the proper comments were added in the main text).
3) In Scheme 2, under the arrows, add "– H+".
Reply: The correction was made.
4) There was no supporting information in the attached files, so I didn't find Figures S1 and S2.
Reply: We made sure with Editorial office that you receive ESI file this time.
5) Line 278-279. "Such phenomenon was previously studied by X-ray crystalography of Cu(II)/metformin complexes in different pH conditions. " There is no reference. Are these the results of the authors?
Reply: Proper references were added (signed in red in the revision version of the manuscript)
6) I suggest to add a small sub-conclusion to each sub-chapter summarizing the obtained results for better understanding by the readers.
Reply: The sub-conclusions were added at the end of 3.1 and 3.2 sections-signed in red in the main text (for the section 3.3 the sub-conclusions were already present).
7) The manuscript lacks information on the IC50.
Reply: Many thanks for this great observation from the reviewer. Basically, the information provided in the manuscript and mistakenly referred to as LC50 represents IC50. A screenshot showing the actual calculation was also included with this letter. Correction of the typo was made accordingly.
8) What mechanism of influence of Cu (II) on the anticancer property of metformin do the authors suggest? How can it be confirmed?
Reply: Different possible mechanisms, also contemporary, could be suggested. Following References 25-29, it is likely that copper/Metf complexes bind to DNA (intercalation mechanism) and could lead to cell cycle arrest and/or mitosis inhibition (to confirm e.g. with fluorescent marker of cell cycles; TEM imaging). It is also likely that copper/Metf complexes bind to biologically important peptides and/or molecules (e.g. glutathione) and inhibit their function, but further specific studies are needed (e.g. immune-, fluorescent staining; mRNA expression and much more). High redox potential of Copper/Metf complex (reference 45) suggest reactive oxygen species (ROS) production as one of the mechanism leading to disruption of nucleic acids and other biological important molecules (studies with different ROS markers are needed to confirm).
Also, several studies indicated that Metf induces apoptosis of cancer cells via targeting of the AMPK and mTOR pathways. This is usually confirmed by exposing the cells to certain factors that inhibits or/and activate the particular pathway and subsequently assess the effects on cells proliferation. For instance, in order to investigate the involvement of AMPK pathway in the effect of Metfromin, in a study by Lu and colleagues human gastric cancer cells were treated with an AMPK inhibitor, compound C and then p-AMPK (indicative of AMPK activation) and cell viability were assessed.
9) I suggest to change "antitumoral" to "anticancer" or "antitumor". In the text of the article, the authors use only 1 term "anticancer".
Reply: Proper correction was made.
10) The manuscript should be rechecked for typos and errors. For example:
Reply: The manuscript was revised for typo errors.
Line 127. "copper sulfate (CuCo4)" should be corrected.
Reply: Proper correction was made.
Figure 6B - "GSH:Cu(I):metformin" should be corrected.
Reply: Proper correction was made.

Reviewer 2 Report
This manuscript reported in-depth the Cu(I) and Cu(II) complexes of metformin and their antitumor activity. First of all, the authors studied the protonation and complex-formation equilibria for metformin by a combined potentiometric-UV titration and 1H-NMR titration at different pH conditions, in particularly in the presence of GSH. Then comprehensive anticancer activity of such compounds have been screened with various tumor types, including cancer cell lines SW1222, K562 and K562-200 (a multi-467 drug resistant form of K562), revealing the anticancer activity of metformin on three cell 468 lines representing two of the most common types of cancers. The findings summarized herein may find useful in future cancer treatments. I suggest the publication of this article after the following corrections are made:
- I would suggest to remove the potentiometric-UV titration and 1H-NMR titration spectra for the complexation studies, as I doubt all these spectroscopic data have already been reported in the literature (ref. 37-43 and those reporting the X-ray structures of Cu(II)/metformin in CCDC).
- Over-fragmented 1H NMR shown in Fig 5 and 6 looks odd. I would suggest to show the stacked full-spectra of the titration.
Author Response
Dear Reviewer,
Together with all authors I would like to thank you for your valuable revision and suggestions regarding our manuscript. Please find in the following brief descriptions of the changes we made in order to improve our paper, in particular with regard to data quality and presentation.
- I would suggest to remove the potentiometric-UV titration and 1H-NMR titration spectra for the complexation studies, as I doubt all these spectroscopic data have already been reported in the literature (ref. 37-43 and those reporting the X-ray structures of Cu(II)/metformin in CCDC).
Reply: It is the first time that the metformin protonation model has 3 labile protons and both potentiometric and 1H-NMR titrations support this hypothesis. Further, the (Metformin)H3 model triggers new metal coordination model and stability constants. For these reasons, we prefer to keep the potentiometric and NMR titration data in the main manuscript.
- Over-fragmented 1H NMR shown in Fig 5 and 6 looks odd. I would suggest to show the stacked full-spectra of the titration.
Reply: Figure 6 was revised properly.
Best regards
Joanna I. Lachowicz

Round 2
Reviewer 1 Report
I thank the authors for correcting the manuscript.
Regarding reply 1. One of the two selected references (Molecules 2021, 26(8), 2334) does not match the list in the manuscript.